# Research on multi-sensor fusion architecture for highway area hazard monitoring based on UAVs

Hui Wu[1,2]*, Shaopeng Li[3], Hua Shan[4], Yi Lu[1], Kaiyuan Hu[5], Wei Zheng[6], Jianbin Xie[7,8]

1 School of Transportation, Changsha University of Science and Technology, Changsha, China, 2 Changsha Youlian Engineering Technology Consulting Co., Ltd., Changsha, China, 3 China West Construction Hunan Co., Ltd., Changsha, China, 4 Information Center of Industry and Information Technology Department of Hunan Province, Changsha, China, 5 School of Computer, University of South China, Hengyang, China, 6 School of Physics and Electronics Science, Changsha University of Science and Technology, Changsha, China, 7 Shandong Xiehe University, Jinan, China, 8 Hunan Zhongke Zhuying Intelligent Technology Research Institute Co., Ltd., Changsha, China

* 850185985@qq.com

## Abstract

Highway area geohazards pose a serious threat to traffic safety, and traditional monitoring echniques struggle to meet the requirements for high precision and real-time capability, while existing unmanned aerial vehicles (UAVs) multi-sensor fusion schemes lack adaptability across diverse areas. Based on the monitoring requirements in different areas and variations in sensor performance, this study employs an improved AHP-TOPSIS method to conduct sensor fusion screening research. Firstly, an improved AHP method was adopted to convert qualitative requirements into quantitative weights; Secondly, integrating the improved TOPSIS method, based on the sensor performance scores after flying height correction and indicator weights, the single-sensor closeness was calculated, thereby screening and forming a single-sensor candidate set; Finally, the TOPSIS process was reapplied for all possible multi-sensor fusion schemes in the candidate set to evaluate these schemes and screen out the optimal fusion scheme for rural level, rural rolling, rural mountainous, and urban/suburban highway areas. Experimental results indicate: For rural level area highways, sensor fusion transitions from basic optical (RGB + IR) to refined optical combinations (RGB/Hyperspectral+Thermal infrared multispectral) as the height of UAVs increases; the rural rolling and mountainous area highways adopt RGB + LiDAR at low-to-medium heights (5-100m), while adjusting to Hyperspectral+SAR and LiDAR + SAR respectively at 100-120m to better adapt to monitoring requirements; urban/suburban area highways use RGB + IR at low-to-medium heights, while RGB+Thermal infrared multispectral proves more suitable at 100-120m heights. This study provides a highly adaptable, precise, and efficient technical scheme for highway hazard monitoring.

**Data availability statement:** All relevant data are within the manuscript.

**Funding:** The author(s) received no specific funding for this work.

**Competing interests:** The authors have declared that no competing interests exist.

## 1. Introduction

As the core component of the transportation network, the safety and stability of highway areas directly impact regional economic operations and public travel safety. Subject to the combined effects of geological structures, climate change (such as extreme rainfall, freeze-thaw cycles), and long-term traffic loads, highway areas are prone to two typical hazards: slope instability and pavement collapse. From the perspective of hazard evolution mechanism, discontinuities such as surface cracks and fracture planes serve as key metrics reflecting slope deformation history and stress state, whose development scale, extension orientation, and distribution density have a significant impact on slope stability [1,2]. Slope instability in the landslide creep stage typically exhibits precursors including slope creeping, fracture development, and groundwater anomalies [3], while the persistent development of pavement cracks may indicate subgrade damage or even collapse risks, serving as a key indicator for hazard precursors and pavement crack assessment [4]. Therefore, the core requirement for precise monitoring of both types of hazards lies in dynamically tracking the evolution of precursor characteristics (e.g., crack propagation), thereby enabling early identification, trend prediction, and precise prevention and control. Although existing UAV technologies have been widely applied in the field of highway hazard monitoring, most monitoring solutions fail to fully adapt to the differentiated demands of complex scenarios, often resulting in low monitoring efficiency or resource waste.

Based on this, this study introduces an improved AHP-TOPSIS method to construct an optimization selection model for UAV multi-sensor fusion solutions tailored to multiple road scenarios, including rural level, rural rolling, rural mountainous, and urban/suburban highway areas, aiming to enhance the scenario adaptability and scientific decision-making of monitoring technologies. The primary objectives of this study are: (1) to develop a methodological framework encompassing sensor performance evaluation indicators, from scenario demand analysis to the optimization of multi-sensor solutions; (2) to validate the applicability and effectiveness of the improved AHP-TOPSIS method in the optimization of sensor fusion solutions; and (3) to identify accurate and efficient sensor combinations suitable for typical road scenarios, including rural level, rural rolling, rural mountainous, and urban/suburban highway areas, for the identification and dynamic tracking of hazard precursors. Nevertheless, this study is subject to certain limitations: sensor performance data is based on industry standards and average literature values, which does not account for performance differences across brands and models; although the model weights were adjusted using a dual-scale method, they remain influenced by expert subjective judgment.

The remainder of this paper is structured as follows: Section 2 Literature Review systematically reviews the current state of research on highway area hazard monitoring technologies, mainstream sensor performance, and sensor fusion; Section 3 Materials and Methods describes the division of research areas, the construction of sensor performance evaluation metrics, the improved AHP weight calculation, and the TOPSIS combined screening process; Section 4 Experimental Analysis and

Results validates the effectiveness of the improved AHP-TOPSIS method and outputs the optimal sensor fusion schemes for each area; Section 5 Conclusions summarizes the research value.

## 2. Literature review

Traditional monitoring methods for highway area hazards mainly include manual inspection, fixed monitoring stations, and satellite remote sensing, among others. In slope monitoring, manual inspection are inefficient and pose safety risks [5], making it difficult to achieve large-scale, high-frequency coverage. Fixed monitoring stations have limited coverage and struggle to adapt to complex terrain. Satellite remote sensing struggles to identify centimeter-level microcracks on slope surfaces due to spatial resolution constraints [6], and its revisit cycles are lengthy (e.g., Sentinel-2 satellites have a 5-day cycle [7]), failing to meet the real-time requirements for early hazard warning. For pavement monitoring, manual inspections are highly subjective [8], with results easily influenced by personnel experience and poor consistency. Specialized inspection vehicles suffer from limited automation and an inability to characterize crack depth [9], rendering them ineffective in detecting concealed distresses such as subgrade voids. Collectively, these traditional approaches exhibit significant constraints in monitoring precision, spatial coverage, and temporal response, thereby leaving them inadequate for the dynamic and high-resolution requirements of highway area hazard monitoring. In recent years, UAV technology has achieved remarkable results in identifying slope instability and pavement cracks due to its advantages of low-height agile flight, rapid deployment, and multi-sensor payloads [10–12]. Yu et al. [13] developed a method for 3D scene reconstruction and hazard identification of highway slopes using UAV oblique photography combined with SfM, MVS, and M3C2 algorithms, successfully detecting a landslide of Heishipu road slope in Changsha, China. Cirillo et al. [14] integrated the PPK method to enhance slope model and rockfall analysis accuracy to below 5 cm. Pan et al. [15] achieved a 98.3% accuracy rate in detecting cracks and potholes using the unmanned aerial vehicle multispectral imagery (UAV MSI) combined with the RF algorithm. Lian et al. [16] used DOM and DEM generated by UAVs to rapidly identify landslides, collapses, and cracks, achieving an accuracy rate of 93% for crack detection and 100% for landslide and collapse detection. In addition, practical applications such as the high slopes along the Tibet's Expressway in China [17] and high-risk slopes at Universiti Malaysia Pahang [18] have demonstrated the significant advantages of UAV technology in enhancing monitoring accuracy and efficiency.

UAVs can be flexibly equipped with a variety of sensorsincluding visible light (RGB) cameras, thermal infrared cameras (Thermal IR Camera), multispectral sensors (Multispectral), hyperspectral sensors (Hyperspectral), Thermal infrared multispectral, Light Detection and Ranging (LiDAR) systems, and Synthetic Aperture Radar (SAR) [19]—to accommodate diverse monitoring requirements, with each offering distinct performance characteristics. The performance characteristics of these sensors vary significantly depending on the specific monitoring area. RGB cameras are used for rock slope stability assessment [20], extracting landslide deformation and surface crack information [21], and identifying pavement defects [22,23]. However, they are highly susceptible to lighting conditions, and is prone to crack information loss due to water film reflections on cloudy or rainy days and backlit environments. And they struggle to penetrate vegetation to capture deep terrain details. IR technology can identify seepage cracks through temperature differences—for instance, post-rain building cracks form thermal "hot/cold zones" due to water infiltration [24]. It also correlates soil moisture levels with regional temperatures [25]. However, its sensitivity is insufficient for non-seepage microcracks (<5mm), and ambient temperature fluctuations may cause false detection. Multispectral sensors derive feature attributes through spectral indices (e.g., Normalized Difference Vegetation Index (NDVI) [26]), aiding in inferring potential triggers for slope failures and geological instability [27]. Hyperspectral sensors, with ultra-high spectral resolution, can capture subtle spectral differences to accurately retrieve the states of rocks, soil, and vegetation, supporting the identification of disaster precursors. However, they are constrained by large volumes of data, complex processing requirements, high costs, and stringent equipment performance requirements. Thermal infrared multispectral sensors combine the advantages of thermal infrared and multispectral technologies, enabling the detection of concealed cracks and water leakage through thermal contrasts, while enhancing

land-cover classification accuracy via multiple spectral bands. This makes them suitable for monitoring hazards in complex environments. LiDAR penetrates vegetation gaps to capture surface information, widely applied in landslide, rockfall, and debris flow monitoring [28]. SAR offers all-weather and all-weather capabilities unaffected by clouds, fog or lighting conditions, has extensive coverage and strong penetration, and is suitable for deformation monitoring (landslides, ground deformation) in rainforest regions [29]. Therefore, A single type of sensor is insufficient to adequately address the diverse monitoring demands, such as slope instability and pavement cracking, whereas multi-sensor fusion enables the effective integration of complementary information, thereby substantially enhancing monitoring robustness. [30]. For instance, the RGB-LiDAR fusion scheme proposed in Reference [31] demonstrated excellent performance in dataset tests, achieving a 4% improvement in recall for crack identification compared to the single RGB scheme, and a 7% increase in segmentation accuracy for sealed cracks—features that lack distinct textural characteristics and are easily overlooked by a single sensor. However, existing fusion schemes predominantly adopt fixed combination patterns and lack a dynamic adaptation mechanism tailored to diverse highway scenarios, resulting in cost redundancy in simple scenarios and limited monitoring efficacy in complex ones. Taking the application of Reference [31] in pavement crack monitoring as an example, its fixed RGB-LiDAR combination neglects scenario variability. In simple scenarios with sufficient lighting and no obstructions (e.g., urban arterial roads), the inclusion of LiDAR not only increases system cost and data processing burden but also offers limited accuracy improvement. Conversely, in complex scenarios with insufficient lighting or dense vegetation, RGB sensors are hindered by vegetation occlusion, and although LiDAR can capture three-dimensional structural information, the overall combination remains susceptible to environmental interference such as rain, fog, and dust, leading to suboptimal monitoring performance.

The Analytic Hierarchy Process – Technique for Order Preference by Similarity to an Ideal Solution (AHP-TOPSIS) method offers the advantages of logical clarity, computational traceability, and efficient ranking [32], providing a scientific approach for the selection of multi-sensor fusion solutions. Its factor weight determination and decision reliability have been validated across multiple fields [33]. However, the traditional AHP method primarily relies on a 1–9 scale to construct judgment matrices, which is prone to subjective bias. Furthermore, existing studies have not fully considered the dynamic impact of factors such as flight height and environmental disturbances on sensor performance, nor have they systematically assessed the adaptability of fusion solutions. This leads to insufficient practicality and reliability of the proposed solutions. Therefore, developing a quantitative adaptation model that integrates scenario demands with sensor performance is key to enhancing the scenario specificity and engineering applicability of multi-sensor fusion solutions.

## 3. Materials and methods

### 3.1. Research areas

Highway road domains exhibit significant variations in topography, environmental conditions, and hazard risks. Consequently, different areas present distinct hazard types (e.g., slope instability probability, pavement distress patterns), precursor characteristics (e.g., micro-crack propagation scale, vegetation anomaly severity), and monitoring frequency requirements. In order to achieve precise alignment between sensor fusion schemes and scene requirements, referring to the classification standards in the Federal Highway Administration (FHWA) *Highway Design Manual* [34], this study classifies highway area into four major areas: (1) rural level, (2) rural rolling, (3) rural mountainous, (4) urban/suburban. The environmental adaptation requirements and core monitoring requirements (precursor feature identification and dynamic evolution tracking) for each area are as follows:

**3.1.1. Environmental adaptation requirements.** Due to differences in terrain, climate, and cultural environment, the four major highway areas exhibit distinct environmental adaptability requirements for sensors. rural level area highways feature stable illumination, with interference primarily from seasonal rain or fog and snow accumulation; electromagnetic interference is weak, vegetation is low-growing, and low penetration requirements are needed for sensors. Rural rolling area highways have good illumination but suffer from localized shrub or scattered tree obstructions; moderate interference

from rain or fog, weak electromagnetic interference, and moderate vegetation penetration requirements necessitate sensors capable of penetrating vegetation to acquire surface information. Rural mountainous highways experience poor illumination, severe mountain obstruction, and frequent rain, fog, and snow; electromagnetic interference is extremely low, vegetation is dense, and high penetration capability is required. Urban/suburban highways have stable illumination, but are obstructed by buildings or roadside trees; rain and fog interference is moderate, electromagnetic interference is strong, thus requiring anti-interference capability and moderate vegetation penetration capability.

**3.1.2. Precursor feature identification.** (1) Precursors to slope instability include microcracks identification, ground deformation, and environmentally correlated features [3]. Microcrack identification requires high spatial resolution sensors to accurately capture the length, orientation, and density of microcracks ≥2 mm wide, while distinguishing active cracks such as seepage fissures [35]. Ground deformation requires precise monitoring of slope displacement, gradient changes, and volume of unstable rock mass. High resolution 3D data (e.g., LiDAR) is the optimal choice, though RGB oblique photography can also assist during routine inspections [13]. Environmental correlation feature monitoring requires sensors capable of capturing surrounding environmental information. For instance, vegetation anomalies can be identified through NDVI multispectral inversion [3]; temperature anomalies can be detected via infrared thermal imaging. Shallow soil temperature anomalies weaken soil strength, while deep soil temperature anomalies indirectly indicate geotechnical sliding risks [36];

(2) Precursors to pavement collapse include surface cracks and deep structural abnormalities. Surface cracks require high spatial resolution sensors to identify transverse and longitudinal cracks with length ≥5 cm and width ≥1 mm; deep structural abnormalities necessitate detection of hidden defects such as subgrade voids and through cracks to ensure pavement structural safety.

**3.1.3. Dynamic evolution tracking.** The quantification of hazard evolution patterns relies on time-series data, necessitating consistent sensor data and cost-effective solutions. The monitoring process requires recording the daily variation of slope cracks and the weekly variation of pavement cracks, with a spatial registration error of ≤3 pixels for multi-temporal data—a requirement dependent on the stable imaging performance of the sensor. Routine inspection frequency can be 1–2 months per visit, with high-risk zones requiring 1–2 weeks per visit. Therefore, sensors must be low-cost and lightweight to accommodate high-frequency deployment. Additionally, high-risk zones require supplementary high-precision sensors to enhance monitoring accuracy.

## 3.2. Construction of sensor performance evaluation metrics

This study established a quantitative sensor performance evaluation system encompassing 14 metrics across three major categories: fundamental performance, environmental adaptability, and precursor feature recognition capability for main-stream sensors—including RGB, Thermal IR Camera, Multispectral, Hyperspectral, Thermal infrared multispectral, LiDAR, and SAR—based on the primary application scenario of urban road crack inspection using UAVs. This system employs a "5-point scale" (higher scores indicate superior performance) to provide quantitative support for sensor selection across diverse areas. Specific scoring results are presented in Table 1.

As shown in Table 1, each sensor exhibits distinct performance characteristics that align clearly with specific application requirements. RGB scores 5 points for spatial resolution, weight, and cost, fully meeting the requirements for lightweight, low-cost, high-resolution crack detection. However, the adaptability to rain/fog/snow conditions scores only 1 point, making it difficult to independently adapt to complex meteorological conditions. IR achieves 5 points for illumination adaptability and 4 points for fog/snow/rain adaptability, making it suitable for complex lighting and weather conditions. Its cost (4 points) is manageable, but its identification of ground deformation (1 point) and cracks (2.5 points) is weak, requiring combination with other sensors to address shortcomings. Multispectral scores 4 points for environmental-related features identification capability, with moderate weight (4 points) and cost (3 points), but its spatial resolution (2 points) is low, making it difficult to detect micro-cracks. Hyperspectral shows outstanding environment-related features identification capability (5 points),

**Table 1. Multi-sensor performance evaluation metric system and corresponding scoring [10,11].**

| Type | Index | Metrics | RGB | IR | Multi-spectral | Hyper-spectral | Thermal infrared multispectral | LiDAR | SAR |
|---|---|---|---|---|---|---|---|---|---|
| **Basic Performance** | $U_1$ | Spatial Resolution | 5 | 3 | 2 | 1.5 | 2.5 | 4 | 1 |
| | $U_2$ | Weight | 5 | 3 | 4 | 2 | 1.5 | 2 | 1 |
| | $U_3$ | Cost | 5 | 4 | 3 | 1.5 | 2 | 2 | 1 |
| **Environmental Adaptability** | $U_4$ | Illumination | 1 | 5 | 2 | 1.5 | 3 | 3 | 4 |
| | $U_5$ | Rain | 1 | 4 | 2 | 1.5 | 3 | 3 | 5 |
| | $U_6$ | Fog | 1 | 4 | 2 | 1.5 | 3 | 3 | 5 |
| | $U_7$ | Snow | 1 | 4 | 2 | 1.5 | 3 | 3 | 5 |
| | $U_8$ | Electromagnetic | 5 | 4 | 3 | 2 | 4 | 2 | 1 |
| | $U_9$ | Vegetation Penetration | 1 | 2 | 2 | 3 | 2 | 4 | 5 |
| **Precursor Identification Capability** | $U_{10}$ | Slope Cracks | 5 | 2 | 3 | 3 | 3 | 4 | 1 |
| | $U_{11}$ | Ground Deformation | 1 | 3 | 2 | 2 | 1.5 | 4 | 5 |
| | $U_{12}$ | Environment-Related Features | 3 | 5 | 4 | 5 | 4 | 2 | 1 |
| | $U_{13}$ | Pavement Cracks | 5 | 3 | 4 | 3.5 | 3 | 2 | 1 |
| | $U_{14}$ | Deep Structural Anomalies | 1 | 3 | 1 | 2 | 3.5 | 4 | 5 |

yet its spatial resolution is low (1.5 points) and cost is relatively high (1.5 points). Thermal infrared multispectral performs in a more balanced manner, achieving good scores in environmental-related features identification capability (4 points) and illumination/rain/fog/snow adaptability (3 points), though its weight and cost are relatively high.LiDAR achieves 4 points for ground deformation and deep structural anomalies monitoring, with strong vegetation penetration (4 points), matching rural rolling and mountainous highway needs for "penetrating vegetation and detecting deep deformation." SAR achieves 5 points for rain/fog/snow adaptability and 5 points for vegetation penetration—the best overall. Its imaging is minimally affected by height, making it suitable for rural rolling and mountainous area highways. However, its electromagnetic interference resistance scores only 1 point (warranting particular caution in the urban area), and both its environment-related features identification and crack detection capabilities score 1 point, necessitating fusion with other sensors.

It should be noted that the sensor performance scores above are based on the average level of typical areas, and the UAV flying height As a critical operational parameter for actual highway monitoring, the UAV flying height further impacts sensor data quality and performance. Therefore, quantitative analysis must incorporate height factors. Research indicates that flight height significantly affects sensor performance: LiDAR's Digital Terrain Model (DTM) accuracy decreases with increasing height, and the error increases more noticeably in complex terrain [37]; At 120 m height, RGB sensors achieve a ground sampling distance of 5.3 cm (compared to only 1.3 cm at 30 m), while IR resolution consistently lags behind RGB at the same height [38]; both RGB and multispectral sensors exhibit a linear negative correlation between ground resolution and height. Multispectral performance remains stable between 25–120 m, whereas RGB shows significant vegetation index variations at 61m height [39]. Therefore, this study classifies flying heights into four intervals: low (5–20 m), medium-low (20–50 m), medium-high (50–100 m), and high (100–120 m). A coefficient of "1.0" is established as the baseline for no performance influence, where lower coefficients indicate a greater negative influence. The specific influence coefficients are provided in Table 2.

According to Table 2, optical sensors exhibit optimal performance at low heights, with resolution degrading at medium-high heights. At high heights, both RGB and IR thermal resolution decline significantly. The point cloud density of LiDAR is inversely proportional to the square of the height, exhibiting the most rapid decline. SAR resolution has no direct correlation with height, maintaining stable imaging quality. The flying height does not alter the sensor's adaptive logic for environmental factors. Only SAR is affected by increased flying height, resulting in heightened electromagnetic interference. The

**Table 2. Influence coefficients of varying UAV flying heights on sensor performance.**

| Height | Index | RGB | IR | Multispectral | Hyperspectral | Thermal infrared multispectral | LiDAR | SAR |
|---|---|---|---|---|---|---|---|---|
| **5-20 m** | $U_1$ | 0.98 | 0.96 | 0.98 | 0.98 | 0.97 | 0.96 | 0.99 |
| | $U_2$ | 1 | 1 | 1 | 1 | 1 | 1 | 1 |
| | $U_3$ | 1 | 1 | 1 | 1 | 1 | 1 | 1 |
| | $U_4$ | 1 | 1 | 1 | 1 | 1 | 1 | 1 |
| | $U_5$ | 1 | 1 | 1 | 1 | 1 | 1 | 1 |
| | $U_6$ | 1 | 1 | 1 | 1 | 1 | 1 | 1 |
| | $U_7$ | 1 | 1 | 1 | 1 | 1 | 1 | 1 |
| | $U_8$ | 1 | 1 | 1 | 1 | 1 | 1 | 0.98 |
| | $U_9$ | 1 | 1 | 1 | 1 | 1 | 1 | 1 |
| | $U_{10}$ | 0.98 | 0.96 | 0.98 | 0.98 | 0.97 | 0.96 | 0.98 |
| | $U_{11}$ | 0.98 | 0.96 | 0.98 | 0.98 | 0.97 | 0.96 | 0.98 |
| | $U_{12}$ | 0.98 | 0.96 | 0.98 | 0.98 | 0.97 | 0.96 | 0.98 |
| | $U_{13}$ | 0.98 | 0.96 | 0.98 | 0.98 | 0.97 | 0.96 | 0.98 |
| | $U_{14}$ | 0.95 | 0.93 | 0.95 | 0.95 | 0.94 | 0.96 | 0.98 |
| **20-50 m** | $U_1$ | 0.93 | 0.90 | 0.96 | 0.96 | 0.93 | 0.88 | 0.98 |
| | $U_2$ | 1 | 1 | 1 | 1 | 1 | 1 | 1 |
| | $U_3$ | 1 | 1 | 1 | 1 | 1 | 1 | 1 |
| | $U_4$ | 1 | 1 | 1 | 1 | 1 | 1 | 1 |
| | $U_5$ | 1 | 1 | 1 | 1 | 1 | 1 | 1 |
| | $U_6$ | 1 | 1 | 1 | 1 | 1 | 1 | 1 |
| | $U_7$ | 1 | 1 | 1 | 1 | 1 | 1 | 1 |
| | $U_8$ | 1 | 1 | 1 | 1 | 1 | 1 | 0.95 |
| | $U_9$ | 1 | 1 | 1 | 1 | 1 | 1 | 1 |
| | $U_{10}$ | 0.93 | 0.90 | 0.96 | 0.96 | 0.93 | 0.88 | 0.98 |
| | $U_{11}$ | 0.9 | 0.90 | 0.96 | 0.96 | 0.93 | 0.9 | 0.98 |
| | $U_{12}$ | 0.93 | 0.90 | 0.96 | 0.96 | 0.93 | 0.9 | 0.98 |
| | $U_{13}$ | 0.93 | 0.90 | 0.96 | 0.96 | 0.93 | 0.88 | 0.98 |
| | $U_{14}$ | 0.93 | 0.90 | 0.89 | 0.89 | 0.90 | 0.88 | 0.98 |
| **50-100 m** | $U_1$ | 0.86 | 0.85 | 0.94 | 0.92 | 0.89 | 0.78 | 0.96 |
| | $U_2$ | 1 | 1 | 1 | 1 | 1 | 1 | 1 |
| | $U_3$ | 1 | 1 | 1 | 1 | 1 | 1 | 1 |
| | $U_4$ | 1 | 1 | 1 | 1 | 1 | 1 | 1 |
| | $U_5$ | 1 | 1 | 1 | 1 | 1 | 1 | 1 |
| | $U_6$ | 1 | 1 | 1 | 1 | 1 | 1 | 1 |
| | $U_7$ | 1 | 1 | 1 | 1 | 1 | 1 | 1 |
| | $U_8$ | 1 | 1 | 1 | 1 | 1 | 1 | 0.9 |
| | $U_9$ | 1 | 1 | 1 | 1 | 1 | 1 | 1 |
| | $U_{10}$ | 0.86 | 0.85 | 0.94 | 0.92 | 0.89 | 0.78 | 0.96 |
| | $U_{11}$ | 0.86 | 0.85 | 0.94 | 0.92 | 0.89 | 0.83 | 0.96 |
| | $U_{12}$ | 0.86 | 0.85 | 0.94 | 0.92 | 0.89 | 0.83 | 0.96 |
| | $U_{13}$ | 0.86 | 0.85 | 0.94 | 0.92 | 0.89 | 0.78 | 0.96 |
| | $U_{14}$ | 0.86 | 0.85 | 0.94 | 0.92 | 0.89 | 0.83 | 0.96 |

*(Continued)*

Table 2. (Continued)

| Height | Index | RGB | IR | Multispectral | Hyperspectral | Thermal infrared multispectral | LiDAR | SAR |
|---|---|---|---|---|---|---|---|---|
| **100-120 m** | $U_1$ | 0.68 | 0.65 | 0.92 | 0.88 | 0.78 | 0.70 | 0.94 |
| | $U_2$ | 1 | 1 | 1 | 1 | 1 | 1 | 1 |
| | $U_3$ | 1 | 1 | 1 | 1 | 1 | 1 | 1 |
| | $U_4$ | 1 | 1 | 1 | 1 | 1 | 1 | 1 |
| | $U_5$ | 1 | 1 | 1 | 1 | 1 | 1 | 1 |
| | $U_6$ | 1 | 1 | 1 | 1 | 1 | 1 | 1 |
| | $U_7$ | 1 | 1 | 1 | 1 | 1 | 1 | 1 |
| | $U_8$ | 1 | 1 | 1 | 1 | 1 | 1 | 0.85 |
| | $U_9$ | 1 | 1 | 1 | 1 | 1 | 1 | 1 |
| | $U_{10}$ | 0.68 | 0.65 | 0.92 | 0.88 | 0.78 | 0.7 | 0.94 |
| | $U_{11}$ | 0.68 | 0.65 | 0.92 | 0.88 | 0.78 | 0.75 | 0.94 |
| | $U_{12}$ | 0.68 | 0.65 | 0.92 | 0.88 | 0.78 | 0.75 | 0.94 |
| | $U_{13}$ | 0.68 | 0.65 | 0.92 | 0.88 | 0.78 | 0.7 | 0.94 |
| | $U_{14}$ | 0.68 | 0.65 | 0.92 | 0.88 | 0.78 | 0.75 | 0.94 |

ability to identify precursor features is strongly correlated with resolution and imaging quality. Precursor identification capability is strongly correlated with resolution and imaging quality. At low heights (5-20m), RGB imagery excels at recognizing subtle features like slope cracks and pavement cracks due to the high resolution. LiDAR maintains consistently high recognition coefficients by leveraging its three-dimensional data advantages. SAR, benefiting from microwave penetration and the stability of interferometry, demonstrates significant advantages in identifying macro-level features such as ground deformation and deep structural anomalies.

After completing data processing for sensor performance evaluation metrics, the subsequent text will apply an improved AHP-TOPSIS method to conduct sensor fusion screening research based on monitoring requirements across different areas and sensor performance variations. The multi-sensor fusion schemes screening process based on improved AHP-TOPSIS is illustrated in Fig 1.

### 3.3. Calculating metric weights based on improved AHP

To achieve precise alignment between sensor performance and highway area requirements, it is necessary to first clarify the core demand metrics of the four major highway areas and establish a one-to-one mapping relationship with the sensor performance evaluation metrics in Table 1, thereby transforming qualitative descriptions into quantitative correlations. The traditional AHP method, relying solely on the 1–9 scale, is prone to bias due to subjective judgments. Therefore, this study employs an improved AHP method that integrates 1–9 scales with $e^{0/4}$ $e^{8/4}$ scales [40]. First, the weights of the 14 metrics are calculated using both scales, and after consistency testing to ensure reliability, the arithmetic mean of the two sets of weights is taken as the final weight. This method enables reasonable weight allocation for the 14 metrics based on area-specific priority requirements (see Table 3 for details), ensuring that sensor performance prioritizes core needs while balancing environmental adaptability and engineering feasibility.

The steps for determining the weight of sensor performance metrics using the improved AHP method are as follows:

**(1) Establish judgment matrices;**

Conduct pairwise comparisons of metric importance based on the 1–9 scale and the $e^{0/4}$ $e^{8/4}$ scale to construct the judgment matrices $A = (a_{ij})_{n \times n}$ where $n$ represents the number of indicators, and $a_{ij}$ denotes the importance scale value of the $i$-th indicator relative to the $j$-th indicator.

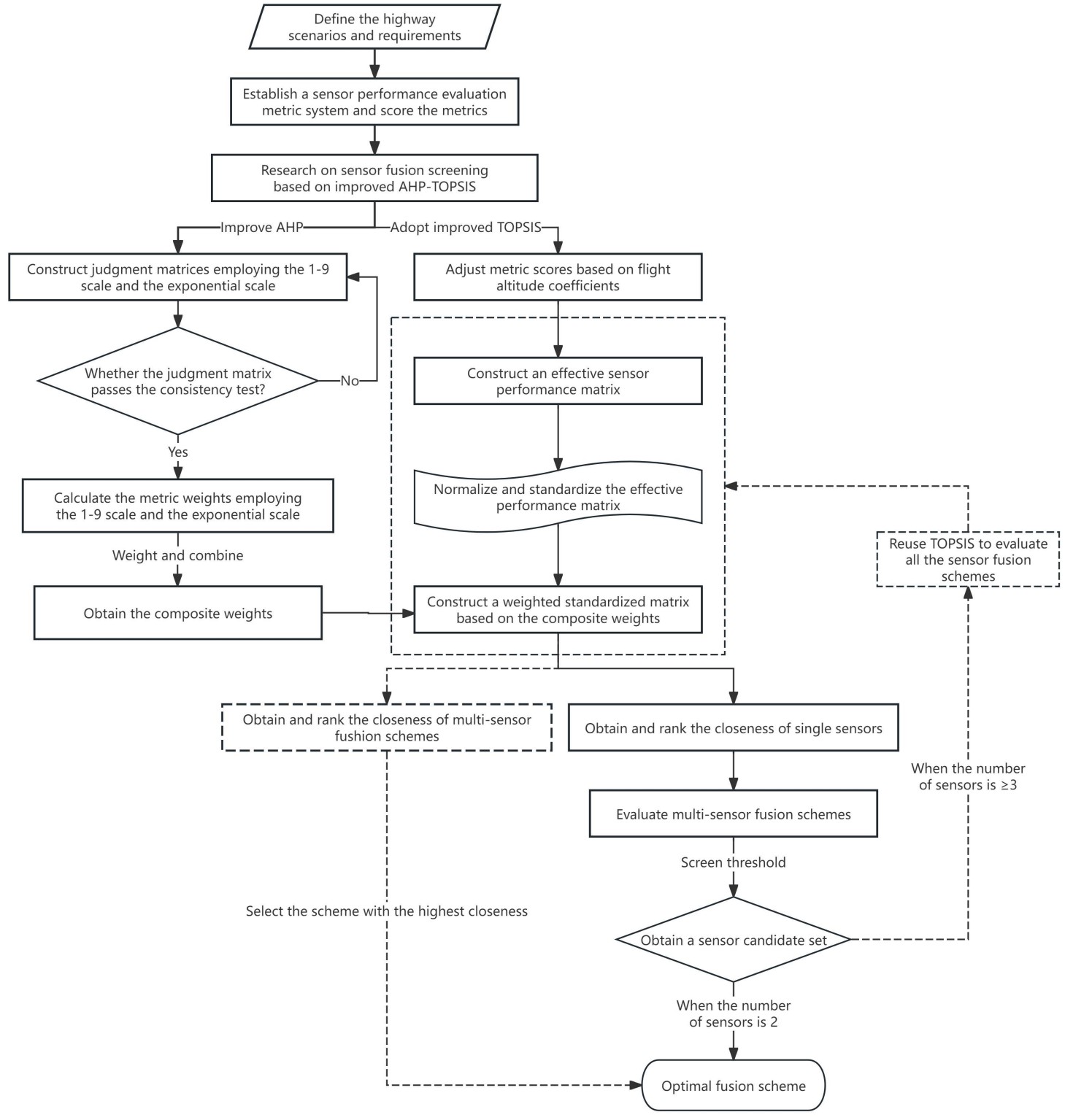

**Fig 1. Flowchart for multi-sensor fusion scheme selection based on improved AHP-TOPSIS.**

Table 3. **Specific requirements for sensor environmental adaptability and hazard response capability in four major areas.**

| Area Type | Environmental Adaptability | Core Requirements |
|---|---|---|
| **Rural Level** | Illumination: Relatively stable;<br>Weather: Seasonal rain/fog/snow accumulation (moderate frequency of disruption, primarily winter snow accumulation);<br>Electromagnetic: Low interference;<br>Vegetation: Low vegetation cover, low penetration capability required | High Priority: Pavement Crack Monitoring (Core), Periodic Inspection of Subgrade Voids;<br>Medium Priority: Seasonal Monitoring of Temperature Anomalies;<br>Low Priority: Routine Slope Monitoring (Low Risk), NDVI Inversion (Weak Demand) |
| **Rural Rolling** | Illumination: Good (local mountain shading, controllable interference);<br>Weather: Moderate rain/fog interference (frequent in rainy season, little snow accumulation);<br>Electromagnetic: Low interference;<br>Vegetation: Shrubs/sparse trees, moderate penetration capability required | High Priority: Precise identification of slope microcracks, ground deformation monitoring, and key monitoring of through cracks in road surfaces;<br>Medium Priority: Simultaneous monitoring of NDVI and temperature anomalies. |
| **Rural Mountainous** | Illumination: Poor (dense mountain shading, short effective daylight duration);<br>Weather: Frequent rain/fog/snow (high annual interference frequency, heavy snow accumulation in winter);<br>Electromagnetic: extremely low interference;<br>Vegetation: High coverage, high penetration capability required | High Priority: Slope microcrack monitoring, ground deformation monitoring, environmental correlation features (vegetation anomalies + temperature anomalies, core);<br>Medium Priority: High-frequency monitoring of pavement cracks and subgrade voids. |
| **Urban/suburban** | Illumination: Stable (but with building/road tree shading);<br>Weather: Moderate rain/fog interference (frequent in rainy season/hazy days, snow easy to remove);<br>Electromagnetic: High interference (dense surrounding base stations and power grids);<br>Vegetation: Road tree distribution, moderate penetration capability required | High Priority: Pavement crack monitoring, deep structural anomaly monitoring (core);<br>Medium Priority: Local unstable rock mass monitoring, ground deformation monitoring, subgrade/pavement temperature anomaly monitoring;<br>Low Priority: Environment-related features (low demand), slope monitoring (limited quantity) |

**(2) Calculate metric weights;**

According to matrix theory, the weight coefficients of each factor correspond to the eigenvectors of the judgment matrix $W$, satisfying the following eigenvector equation:

$$AW = \lambda_{max} W, \tag{1}$$

where $A$ is the judgment matrix, $\lambda_{max}$ is the maximum eigenvalue of the matrix, and $W$ is the eigenvector to be calculated (i.e., the indicator weight vector). Common methods for calculating eigenvectors include the square root method, power method, arithmetic mean method, and geometric mean method. This paper employs the geometric mean method for calculation, with the specific steps outlined as follows:

1) Normalize the judgment matrix columns

For each column element of the judgment matrix $A = (a_{ij})_{n \times n}$ (where $n$ is the number of metrics), perform column summation followed by normalization to obtain the column-normalized matrix:

$$b_{ij} = a_{ij} / \sum_{k=1}^{n} a_{kj}, \tag{2}$$

2) Sum rows to obtain initial weights

For the column-normalized matrix, accumulate the sum row-wise to obtain the preliminary weight of the $i$-th indicator:

$$w_i^{(0)} = \sum_{j=1}^{n} b_{ij}, \tag{3}$$

3) Normalize to obtain final weights

The initial weight vector $w_i^{(0)}$ is further normalized such that the sum of all indicator weights equals 1, yielding the final weight vector: $W = (w_1, w_2, \cdots, w_n)^T$. The equation is as follows:

$$w_i = w_i^{(0)}/\sum_{k=1}^{n} w_k^{(0)}, \tag{4}$$

4) Calculate the maximum eigenvalue $\lambda_{max}$.

$$\lambda_{max} = \frac{1}{n}\sum_{i=1}^{n} \frac{(AW)_i}{w_i}, \tag{5}$$

where $(AW)_i$ denotes the $i$-th component of $AW$.

**(3) Consistency check;**

Due to the subjectivity of the importance of metrics, consistency check of the judgment matrix is required. The checking steps are as follows:

1) Calculate consistency metrics.

$$CI = (\lambda_{max} - n)/(n - 1), \tag{6}$$

2) Calculate consistency ratio.

$$CR = CI/RI, \tag{7}$$

where $RI$ is the average random consistency index, whose value can be obtained by consulting Table 4. If $CR < 0.1$, the differences in the judgment matrix are considered within the permissible range and have consistency.

**(4) Obtain composite weight vector.**

By weighting, the weights $W_1$ and $W_2$ are combined to obtain the composite weight vector $W$, where $W = \frac{(W_1 + W_2)}{2}$, $W_1$ is the weight value obtained using the 1–9 scale, and $W_2$ is the weight value obtained using the $e^{0/4}$ $e^{8/4}$ scale.

### 3.4. Screening multi-sensor fusion based on improved TOPSIS

(1) Construct an effective sensor performance matrix;

Sensors performance degrades with increasing flying height. The original performance score must be adjusted using the flying height coefficient to obtain the actual effective score $x_{ij}$ for a specific area at a specific height. The equation is:

$$x_{ij} = x'_{ij} \cdot f(h), \tag{8}$$

where $x'_{ij}$ represents the score of the $i$-th sensor under the $j$-th performance metric, and $f(h)$ represents the influence coefficient corresponding to the flying height $f(h)$ (Table 2).

Construct an effective performance matrix $X = (x_{ij})_{m \times n}$ based on actual effective scores $x_{ij}$. Here, $m = 7$ is the number of sensor types; $n = 14$ represents the number of sensor performance metrics.

**Table 4. Random consistency check table.**

| $n$ | 1 | 2 | 3 | 4 | 5 | 6 | 7 | 8 | 9 | 10 | 11 | 12 | 13 | 14 | 15 |
|---|---|---|---|---|---|---|---|---|---|---|---|---|---|---|---|
| $RI$ | 0 | 0 | 0.52 | 0.89 | 1.12 | 1.26 | 1.36 | 1.41 | 1.46 | 1.49 | 1.52 | 1.54 | 1.56 | 1.58 | 1.59 |

**(2) Perform normalization processing;**

TOPSIS requires all metrics to be positive metrics (where higher values indicate superior performance). Since the sensor performance indicators in Table 1 already satisfy the characteristic that "higher scores indicate better performance" (e.g., cost score, weight score, etc.), they are all positive indicators and thus require no additional normalization.

**(3) Standardize a effective performance matric;**

To eliminate the impact of differences in metric magnitude on the calculation results, the effective performance matrix $X$ is normalized using the vector normalization method, yielding the normalized matrix $Z = (z_{ij})_{m \times n}$. The equation is as follows:

$$z_{ij} = \frac{x_{ij}}{\sqrt{\sum_{i=1}^{m} x_{ij}^2}} \qquad (i = 1, 2, \cdots, m; j = 1, 2, \cdots, n),$$

(9)

where $x_{ij}$ is the effective score of the $i$-th sensor under the $j$-th performance metric.

**(4) Construct a weighted standardization matrix;**

Combine the standardized matrix $Z$ with the weight vector $W$ to obtain a weighted standardized matrix $V = (v_{ij})_{m \times n}$. The calculation equation is:

$$v_{ij} = z_{ij} \times w_j \qquad (i = 1, 2, \cdots, m; j = 1, 2, \cdots, n),$$

(10)

where $v_{ij}$ represents the standardized score of the $i$-th sensor under the $j$-th performance metric after weight adjustment, embodying the influence of the discrepancy in metric importance on the comprehensive performance.

**(5) Calculate and rank closeness and rank**

1) Determine positive and negative ideal solutions

Since all metrics are positive metrics (higher values indicate better performance), therefore:

The positive ideal solution $PIS^+$ is a vector composed of the maximum values of each metric, i.e.,

$$PSI^+ = \left( v_1^+, v_2^+, \cdots, v_n^+ \right), \quad v_j^+ = \max_{i=1}^{m} v_{ij},$$

(11)

The negative ideal solution $PIS^-$ is a vector composed of the minimum values of each metric, i.e.,

$$PSI^- = \left( v_1^-, v_2^-, \cdots, v_n^- \right), \quad v_j^- = \min_{i=1}^{m} v_{ij},$$

(12)

2) Calculate the Euclidean distance from each sensor to the ideal solution

Distance to positive ideal solution $D_i^+$:

$$D_i^+ = \sqrt{\sum_{j=1}^{n} (v_{ij} - v_j^+)^2} \qquad (i = 1, 2, \cdots, m),$$

(13)

Distance to negative ideal solution $D_i^-$:

$$D_i^- = \sqrt{\sum_{j=1}^{n} (v_{ij} - v_j^-)^2} \qquad (i = 1, 2, \cdots, m),$$

(14)

 

3) Calculate closeness

The closeness $C_i$ of the $i$-th sensor is defined as:

$$C_i = \frac{D_i^-}{D_i^+ + D_i^-} \qquad (1, 2, \cdots, m),$$

(15)

$C_i \in [0, 1]$, the closer $C_i$ is to 1, the closer the sensor is to the positive ideal solution and the farther it is form negative ideal solution, indicating better overall performance. Sensors are ranked in descending order of $C_i$ to determine their relative performance.

**(6) Evaluate multi-sensor fusion schemes;**

1) Screen and form a single-sensor candidate set

Calculate the single-sensor candidate threshold $\overline{C} = \frac{\sum_{i=1}^{m} C_i}{m}$ (where $C_i$ is the closeness of the $i$-th sensor and $m$ is the number of sensors), retain sensors with $C_i \geq \overline{C}$ as candidates, and exclude sensors with low closeness to reduce computational load in subsequent multi-sensor fusion. The candidate sensors are denoted as $\{C_1, C_2, \cdots C_k\}$ ($k \leq m$).

2) Evaluate multi-sensor fusion schemes reusing TOPSIS

When the number of candidate sensors is 2, the combination scheme is directly selected as the optimal solution. If the number of candidate sensors is greater than or equal to 3, the generated multi-sensor fusion schemes must be evaluated. The specific steps are as follows:

Construct a combined effective performance matrix. For each combination scheme, comprehensively consider the performance metrics of each sensor in the combination, calculate the average effective performance score of the combination scheme on each metric, and thereby construct the combination effective performance matrix.

Standardization and weighting. Similar to the single-sensor screening process, the combined effective performance matrix is standardized to eliminate the impact of metric scale differences. Subsequently, incorporating metric weights determined through methods like the improved AHP, a weighted standardized matrix is derived.

Calculate the positive and negative ideal solutions and their closeness. Determine the positive and negative ideal solutions of the combination scheme, calculate the Euclidean distance between each combination scheme and the positive and negative ideal solutions, and then obtain the comprehensive closeness. A higher proximity score indicates that the combination scheme is closer to the ideal sensor fusion and exhibits superior performance.

**(7) Screen the scheme.**

Rank the multi-sensor fusion schemes based on their closeness, and select the scheme with the highest closeness as the optimal sensor fusion scheme.

### 3.5. Extensibility description of research areas

The four aforementioned scenarios—rural level, rural rolling, rural mountainous, and urban/suburban highway areas—are representative of typical highway road scenarios. However, for special highway areas such as permafrost regions and coastal saline zones, the optimal sensor combinations may differ due to the distinct characteristics of their geological conditions (e.g., freeze–thaw cycles in permafrost, corrosion in saline soils), climatic environments (e.g., extremely low temperatures, high humidity), and hazard mechanisms (e.g., thaw settlement deformation, subgrade corrosion). Based on the quantitative adaptation framework of scenario requirements and sensor performance, as well as the improved AHP-TOPSIS method developedin this study, the research results can be extended to such special scenarios systematically and efficiently. This extension mainly follows a three-step approach.:

Step 1: Demand analysis and feature extraction. Identify the core hazard precursor features in the new scenario (e.g., thermokarst cracks in permafrost regions, salt-induced cracks in coastal areas) and their monitoring priorities.

Step 2: Adaptive adjustment of the sensor performance evaluation system. Based on the existing general sensor performance evaluation system, add specialized metrics (e.g., "low-temperature operational stability", "corrosion resistance")

or modify metric weights (e.g., increasing the weights of "snow adaptability" and "ground deformation monitoring capability" in permafrost regions).

Step 3: Reuse of the decision-making process and generation of schemes based on the improved AHP-TOPSIS method. By inputting the updated indicators and weights into the "demand quantification–performance scoring–solution optimization" process established in this study, the optimal sensor fusion solution adapted to the new scenario can be obtained.

## 4. Experimental analysis and results

### 4.1. Experimental tools

This experiment employed the SPSSAU data analysis platform (https://www.spssau.com) to conduct quantitative calculations using the improved AHP-TOPSIS method.

### 4.2. Improve AHP to determine evaluation metric weights

For the four major highway areas, the multi-sensor performance evaluation metric system constructed based on Table 1 was employed to assign weights to the 14 metrics using improved AHP. 3–5 experts were invited in highway hazard monitoring to sequentially compare each pair of the 14 indicators according to the scaling rules shown in Table 5, and obtain judgment matrices $A$ and $A^*$ corresponding to the 1–9 scales and the $e^{0/4}$ $e^{8/4}$ scale. Taking the rural level highway area as an example, the judgment matrices $A$ and $A^*$ constructed after expert assessment are as follows:

$$A_1 = \begin{bmatrix} 1 & 3 & 3 & 1/3 & 1/5 & 1/5 & 1/5 & 3 & 1/3 & 1/3 & 1/3 & 1/3 & 1/9 & 1/7 \\ 1/3 & 1 & 1 & 1/5 & 1/7 & 1/7 & 1/7 & 1 & 1/5 & 1/5 & 1/5 & 1/5 & 1/9 & 1/7 \\ 1/3 & 1 & 1 & 1/5 & 1/7 & 1/7 & 1/7 & 1 & 1/5 & 1/5 & 1/5 & 1/5 & 1/9 & 1/7 \\ 3 & 5 & 5 & 1 & 1/3 & 1/3 & 1/3 & 5 & 3 & 1/3 & 1/3 & 3 & 1/7 & 1/5 \\ 5 & 7 & 7 & 3 & 1 & 1 & 1 & 7 & 5 & 3 & 3 & 5 & 1/5 & 1/3 \\ 5 & 7 & 7 & 3 & 1 & 1 & 1 & 7 & 5 & 3 & 3 & 5 & 1/5 & 1/3 \\ 5 & 7 & 7 & 3 & 1 & 1 & 1 & 7 & 5 & 3 & 3 & 5 & 1/5 & 1/3 \\ 1/3 & 1 & 1 & 1/5 & 1/7 & 1/7 & 1/7 & 1 & 1/5 & 1/5 & 1/5 & 1/5 & 1/9 & 1/7 \\ 3 & 5 & 5 & 1/3 & 1/5 & 1/5 & 1/5 & 5 & 1 & 1/3 & 1/3 & 1 & 1/9 & 1/7 \\ 3 & 5 & 5 & 3 & 1/3 & 1/3 & 1/3 & 5 & 3 & 1 & 1 & 3 & 1/7 & 1/5 \\ 3 & 5 & 5 & 3 & 1/3 & 1/3 & 1/3 & 5 & 3 & 1 & 1 & 3 & 1/7 & 1/5 \\ 3 & 5 & 5 & 1/3 & 1/5 & 1/5 & 1/5 & 5 & 1 & 1/3 & 1/3 & 1 & 1/9 & 1/7 \\ 9 & 9 & 9 & 7 & 5 & 5 & 5 & 9 & 9 & 7 & 7 & 9 & 1 & 3 \\ 7 & 7 & 7 & 5 & 3 & 3 & 3 & 7 & 7 & 5 & 5 & 7 & 1/3 & 1 \end{bmatrix}$$

**Table 5. Scale and meaning of two judgment matrices.**

| Importance Level | the 1–9 Scale | The $e^{0/4} \sim e^{8/4}$ scale |
|---|---|---|
| $a_i$ and $a_j$ are of equal importance | 1 | 1 |
| $a_i$ is moderately more important than $a_j$ | 3 | 1.649 |
| $a_i$ is strongly more important than $a_j$ | 5 | 2.718 |
| $a_i$ is very strongly more important than $a_j$ | 7 | 4.482 |
| $a_i$ is extremely more important than $a_j$ | 9 | 7.390 |
| Intermediate values between two adjacent judgments | 2, 4, 6, 8 | 1.284, 2.117, 3.490, 5.755 |

$$A_1^* = \begin{bmatrix}
1 & 1.649 & 1.649 & 1/1.649 & 1/2.718 & 1/2.718 & 1/2.718 & 1.649 & 1/1.649 & 1/1.649 & 1/1.649 & 1/1.649 & 1/7.390 & 1/4.482 \\
1/1.649 & 1 & 1 & 1/2.718 & 1/4.482 & 1/4.482 & 1/4.482 & 1 & 1/2.718 & 1/2.718 & 1/2.718 & 1/2.718 & 1/7.390 & 1/4.482 \\
1/1.649 & 1 & 1 & 1/2.718 & 1/4.482 & 1/4.482 & 1/4.482 & 1 & 1/2.718 & 1/2.718 & 1/2.718 & 1/2.718 & 1/7.390 & 1/4.482 \\
1.649 & 2.718 & 2.718 & 1 & 1/1.649 & 1/1.649 & 1/1.649 & 1.649 & 1/1.649 & 1/1.649 & 1/1.649 & 1.649 & 1/4.482 & 1/2.718 \\
2.718 & 4.482 & 4.482 & 1.649 & 1 & 1 & 1 & 4.482 & 2.718 & 1.649 & 1.649 & 2.718 & 1/2.718 & 1/1.649 \\
2.718 & 4.482 & 4.482 & 1.649 & 1 & 1 & 1 & 4.482 & 2.718 & 1.649 & 1.649 & 2.718 & 1/2.718 & 1/1.649 \\
2.718 & 4.482 & 4.482 & 1.649 & 1 & 1 & 1 & 4.482 & 2.718 & 1.649 & 1.649 & 2.718 & 1/2.718 & 1/1.649 \\
1/1.649 & 1 & 1 & 1/2.718 & 1/4.482 & 1/4.482 & 1/4.482 & 1 & 1/2.718 & 1/2.718 & 1/2.718 & 1/2.718 & 1/7.390 & 1/4.482 \\
1.649 & 2.718 & 2.718 & 1/1.649 & 1/2.718 & 1/2.718 & 1/2.718 & 2.718 & 1 & 1/1.649 & 1/1.649 & 1 & 1/7.390 & 1/4.482 \\
1.649 & 2.718 & 2.718 & 1.649 & 1/1.649 & 1/1.649 & 1/1.649 & 2.718 & 1.649 & 1 & 1 & 1.649 & 1/4.482 & 1/2.718 \\
1.649 & 2.718 & 2.718 & 1.649 & 1/1.649 & 1/1.649 & 1/1.649 & 2.718 & 1.649 & 1 & 1 & 1.649 & 1/4.482 & 1/2.718 \\
1.649 & 2.718 & 2.718 & 1/1.649 & 1/2.718 & 1/2.718 & 1/2.718 & 2.718 & 1 & 1/1.649 & 1/1.649 & 1 & 1/7.390 & 1/4.482 \\
7.390 & 7.390 & 7.390 & 4.482 & 2.718 & 2.718 & 2.718 & 7.390 & 7.390 & 4.482 & 4.482 & 7.390 & 1 & 1.649 \\
4.482 & 4.482 & 4.482 & 2.718 & 1.649 & 1.649 & 1.649 & 4.482 & 4.482 & 2.718 & 2.718 & 4.482 & 1/1.649 & 1
\end{bmatrix}$$

Similarly, judgment matrices were obtained for rural rolling, rural mountainous, and urban/suburban highway areas separately. The metrics weights calculated based on the 1–9 scale and the $e^{0/4}$ $e^{8/4}$ scale, as well as their improved fusion weights, are shown in Table 6.

A consistency test on the judgment matrices was conducted according to formulas (6) to (7). The average random consistency index corresponding to the sample size in this study showed that all judgment matrices are consistency-valid. Therefore, the metric weight vectors $W_l$, $W_r$, $W_m$, $W_u$ for the four major highway areas -level, rolling, mountainous, urban/suburban highway areas – are as follows:

$$W_l = (0.027, 0.017, 0.017, 0.050, 0.094, 0.094, 0.094, 0.017, 0.037, 0.057, 0.057, 0.037, 0.249, 0.152)$$

$$W_r = (0.060, 0.013, 0.012, 0.034, 0.057, 0.057, 0.032, 0.024, 0.050, 0.160, 0.148, 0.112, 0.140, 0.101)$$

$$W_m = (0.030, 0.020, 0.011, 0.053, 0.057, 0.053, 0.057, 0.015, 0.065, 0.180, 0.191, 0.113, 0.077, 0.077)$$

**Table 6. Improved AHP evaluation metric weights table.**

| Metrics | Rural Level | | | Rural Rolling | | | Rural Mountainous | | | Urban/suburban | | |
|---|---|---|---|---|---|---|---|---|---|---|---|---|
| | $W_1$ | $W_2$ | $W$ | $W_1$ | $W_2$ | $W$ | $W_1$ | $W_2$ | $W$ | $W_1$ | $W_2$ | $W$ |
| $U_1$ | 0.022 | 0.032 | 0.027 | 0.059 | 0.061 | 0.060 | 0.026 | 0.035 | 0.030 | 0.104 | 0.092 | 0.098 |
| $U_2$ | 0.012 | 0.021 | 0.017 | 0.011 | 0.015 | 0.013 | 0.017 | 0.024 | 0.020 | 0.081 | 0.078 | 0.080 |
| $U_3$ | 0.012 | 0.021 | 0.017 | 0.009 | 0.014 | 0.012 | 0.009 | 0.014 | 0.011 | 0.081 | 0.078 | 0.080 |
| $U_4$ | 0.046 | 0.053 | 0.050 | 0.028 | 0.040 | 0.034 | 0.047 | 0.060 | 0.053 | 0.014 | 0.026 | 0.021 |
| $U_5$ | 0.095 | 0.093 | 0.094 | 0.050 | 0.063 | 0.057 | 0.050 | 0.064 | 0.057 | 0.045 | 0.056 | 0.051 |
| $U_6$ | 0.095 | 0.093 | 0.094 | 0.050 | 0.063 | 0.057 | 0.047 | 0.060 | 0.053 | 0.045 | 0.056 | 0.051 |
| $U_7$ | 0.095 | 0.093 | 0.094 | 0.025 | 0.039 | 0.032 | 0.050 | 0.064 | 0.057 | 0.013 | 0.022 | 0.018 |
| $U_8$ | 0.012 | 0.021 | 0.017 | 0.021 | 0.028 | 0.024 | 0.012 | 0.018 | 0.015 | 0.083 | 0.083 | 0.083 |
| $U_9$ | 0.034 | 0.041 | 0.037 | 0.046 | 0.055 | 0.050 | 0.062 | 0.068 | 0.065 | 0.078 | 0.077 | 0.078 |
| $U_{10}$ | 0.054 | 0.059 | 0.057 | 0.178 | 0.141 | 0.160 | 0.197 | 0.164 | 0.180 | 0.026 | 0.037 | 0.031 |
| $U_{11}$ | 0.054 | 0.059 | 0.057 | 0.158 | 0.138 | 0.148 | 0.205 | 0.177 | 0.191 | 0.045 | 0.057 | 0.051 |
| $U_{12}$ | 0.034 | 0.041 | 0.037 | 0.114 | 0.111 | 0.112 | 0.118 | 0.107 | 0.113 | 0.014 | 0.020 | 0.017 |
| $U_{13}$ | 0.268 | 0.230 | 0.249 | 0.150 | 0.129 | 0.140 | 0.080 | 0.073 | 0.077 | 0.233 | 0.185 | 0.209 |
| $U_{14}$ | 0.165 | 0.140 | 0.152 | 0.101 | 0.102 | 0.101 | 0.080 | 0.073 | 0.077 | 0.137 | 0.126 | 0.132 |

$$W_u = (0.098, 0.080, 0.080, 0.021, 0.051, 0.051, 0.018, 0.083, 0.078, 0.031, 0.051, 0.017, 0.209, 0.132)$$

(All meet $\sum w_i = 1$).

The results of index weight allocation in diverse areas are shown in Fig 2. As can be seen from Fig 2, monitoring requirements differ across diverse areas, and the assigned index weights also vary accordingly.

### 4.3. Screen multi-sensor fusion schemes based on improved TOPSIS

Taking the level highway area as an example, according to equations (8) to (10), for the area with a flying height of 5–20 m, an effective performance matrix $X = (x_{ij})_{m \times n}$ (where $m = 7$ represents the number of sensor types and $n = 14$ represents the number of performance indicators) was constructed. The vector normalization method was applied to standardize the effective performance matrix $X = (x_{ij})_{m \times n}$, yielding the standardized matrix $Z$. Subsequently, the standardized matrix $Z$ was combined with the weight vector $W$ obtained from improved AHP to produce the weighted standardized matrix $V = (v_{ij})_{m \times n}$, as shown below:

$$V = \begin{bmatrix} 0.017 & 0.011 & 0.011 & 0.006 & 0.012 & 0.012 & 0.012 & 0.010 & 0.005 & 0.033 & 0.022 & 0.012 & 0.153 & 0.018 \\ 0.010 & 0.006 & 0.009 & 0.030 & 0.047 & 0.047 & 0.047 & 0.007 & 0.009 & 0.016 & 0.007 & 0.016 & 0.075 & 0.061 \\ 0.007 & 0.009 & 0.006 & 0.012 & 0.024 & 0.024 & 0.024 & 0.004 & 0.012 & 0.020 & 0.011 & 0.016 & 0.092 & 0.027 \\ 0.005 & 0.004 & 0.003 & 0.009 & 0.018 & 0.018 & 0.018 & 0.004 & 0.014 & 0.020 & 0.015 & 0.020 & 0.107 & 0.035 \\ 0.009 & 0.003 & 0.004 & 0.018 & 0.036 & 0.036 & 0.036 & 0.008 & 0.009 & 0.020 & 0.011 & 0.016 & 0.091 & 0.061 \\ 0.014 & 0.005 & 0.005 & 0.018 & 0.030 & 0.030 & 0.030 & 0.005 & 0.019 & 0.026 & 0.029 & 0.008 & 0.060 & 0.072 \\ 0.003 & 0.002 & 0.002 & 0.024 & 0.059 & 0.059 & 0.059 & 0.002 & 0.023 & 0.007 & 0.037 & 0.004 & 0.031 & 0.091 \end{bmatrix}$$

Using the SPSSAU platform, the positive and negative ideal solutions were calculated. The Euclidean distance between each sensor and the ideal solution was measured, and the closeness was ultimately calculated. The results are shown in Table 7.

Similarly, distance to ideal solution and closeness results for each sensor under different flying heights were calculated for the four major highway areas. The sensors were then ranked in descending order based on $C_i$, yielding a comprehensive performance hierarchy as shown in Table 8.

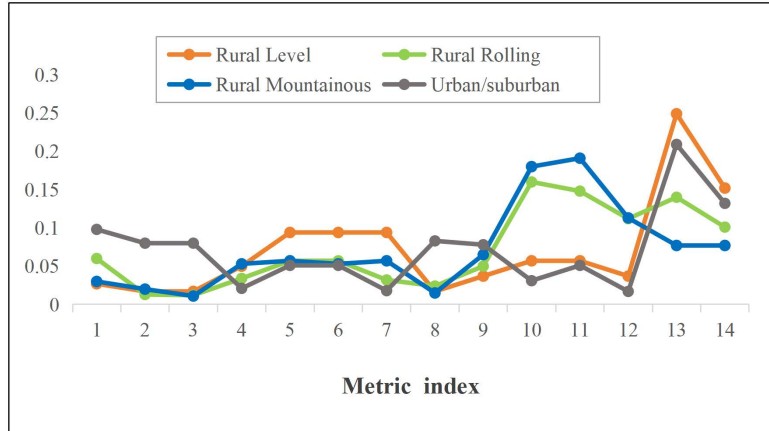

**Fig 2. Comparison of evaluation metric weights in diverse areas.**

**Table 7. Ideal solution distance and closeness results of each sensor in the rural level highway area (flying height 5-20 m).**

| Sensor | Distance to Positive Ideal Solution | Distance to Negative Ideal Solution | Closenessα |
|---|---|---|---|
| **RGB** | 0.116 | 0.128 | 0.526 |
| **IR thermal** | 0.095 | 0.093 | 0.495 |
| **Multispectral** | 0.115 | 0.069 | 0.375 |
| **Hyperspectral** | 0.109 | 0.083 | 0.432 |
| **Thermal infrared multispectral** | 0.089 | 0.088 | 0.499 |
| **LiDAR** | 0.111 | 0.078 | 0.413 |
| **SAR** | 0.128 | 0.117 | 0.477 |

Based on the single-sensor evaluation results in Table 8, first calculate the single-sensor candidate threshold $\overline{C}$. Only sensors with $C_i \geq \overline{C}$ were retained as candidates. After screening, the candidate sensor sets for for the four major areas at different flying heights are shown in Table 9. When the number of candidate sensors is 2, the combination of these 2

**Table 8. Closeness and ranking of sensors across four areas at different heights.**

| Area Type | Flying Height | Ranking of Comprehensive Performance |
|---|---|---|
| **Rural Level** | 5-20m | RGB(0.526)> Thermal infrared multispectral(0.499)> IR thermal(0.495)> SAR(0.477)> Hyperspectral(0.432)> LiDAR(0.413)> Multispectral(0.375) |
| | 20-50m | RGB(0.515)> Thermal infrared multispectral(0.494)> SAR(0.488)> IR thermal(0.486)> Hyperspectral(0.433)> LiDAR(0.392)> Multispectral(0.375) |
| | 50-100m | Thermal infrared multispectral(0.505)> RGB(0.504)> SAR(0.499)> IR thermal(0.487)> Hyperspectral(0.442)> Multispectral(0.393)> LiDAR(0.379) |
| | 100-120m | SAR(0.553)> Thermal infrared multispectral(0.507)> Hyperspectral(0.470)> IR thermal(0.448)> RGB(0.447)> Multispectral(0.433)> LiDAR(0.402) |
| **Rural Rolling** | 5-20m | RGB(0.577)> LiDAR(0.549)> SAR(0.473)> Hyperspectral(0.450)> Thermal infrared multispectral(0.437)> IR thermal(0.399)> Multispectral(0.384) |
| | 20-50m | RGB(0.558)> LiDAR(0.522)> SAR(0.485)> Hyperspectral(0.450)> Thermal infrared multispectral(0.426)> IR thermal(0.384)> Multispectral(0.383) |
| | 50-100m | RGB(0.545)> SAR(0.497)> LiDAR(0.496)> Hyperspectral(0.453)> Thermal infrared multispectral(0.429)> Multispectral(0.397)> IR thermal(0.378) |
| | 100-120m | SAR (0.545)> LiDAR (0.503)> Hyperspectral (0.480)> RGB (0.470)> Multispectral (0.432)> Thermal infrared multispectral (0.413)> IR thermal (0.320) |
| **Rural Mountainous** | 5-20m | LiDAR(0.606)> RGB(0.543)> SAR(0.537)> Hyperspectral(0.417)> Thermal infrared multispectral(0.392)> IR thermal(0.369)> Multispectral(0.356) |
| | 20-50m | LiDAR(0.576)> SAR(0.548)> RGB(0.521)> Hyperspectral(0.416)> Thermal infrared multispectral(0.381)> Multispectral(0.355)> IR thermal(0.354) |
| | 50-100m | SAR(0.559)> LiDAR(0.546)> RGB(0.509)> Hyperspectral(0.416)> Thermal infrared multispectral(0.380)> Multispectral(0.365)> IR thermal(0.348) |
| | 100-120m | SAR (0.604)> LiDAR (0.539)> Hyperspectral (0.437)> RGB (0.433)> Multispectral (0.392)> Thermal infrared multispectral (0.360)> IR thermal (0.299) |
| **Urban/ suburban** | 5-20m | RGB(0.614)> IR thermal(0.468)> Thermal infrared multispectral(0.456)> LiDAR(0.453)> Hyperspectral(0.410)> Multispectral(0.399)> SAR(0.394) |
| | 20-50m | RGB(0.604)> IR thermal(0.460)> Thermal infrared multispectral(0.452)> LiDAR(0.434)> Hyperspectral(0.411)> SAR(0.404)> Multispectral(0.399) |
| | 50-100m | RGB(0.596)> Thermal infrared multispectral(0.462)> IR thermal(0.461)> LiDAR(0.420)> Hyperspectral(0.419)> Multispectral(0.415)> SAR(0.413) |
| | 100-120m | RGB (0.550)> Thermal infrared multispectral (0.465)> SAR (0.457)> Multispectral (0.454)> Hyperspectral (0.450)> LiDAR (0.445)> IR thermal (0.421) |

**Table 9. Screening results of sensor fusion solutions for four major areas at different flight heights.**

| Area Type | Height | Candidate Sensor Set | Combination Scheme | Closeness | Optimal Scheme |
|---|---|---|---|---|---|
| **Rural Level** | 5-20m | RGB、IR thermal、Thermal infrared multispectral、SAR | RGB+IR thermal | 0.549 | RGB+IR |
| | | | RGB+Thermal infrared multispectral | 0.548 | |
| | | | RGB+SAR | 0.520 | |
| | | | IR thermal+Thermal infrared multispectral | 0.480 | |
| | | | IR thermal+SAR | 0.452 | |
| | | | Thermal infrared multispectral+SAR | 0.451 | |
| | | | RGB+IR thermal+Thermal infrared multispectral | 0.538 | |
| | | | RGB+IR thermal+SAR | 0.502 | |
| | | | IR thermal+Thermal infrared multispectral+SAR | 0.454 | |
| | | | RGB+IR thermal+Thermal infrared multispectral+SAR | 0.500 | |
| | 20-50m | RGB、IR thermal、Thermal infrared multispectral、SAR | RGB+IR thermal | 0.541 | RGB+Thermal infrared multispectral |
| | | | RGB+Thermal infrared multispectral | 0.545 | |
| | | | RGB+SAR | 0.526 | |
| | | | IR thermal+Thermal infrared multispectral | 0.474 | |
| | | | IR thermal+SAR | 0.455 | |
| | | | Thermal infrared multispectral+SAR | 0.459 | |
| | | | RGB+IR thermal+Thermal infrared multispectral | 0.531 | |
| | | | RGB+IR thermal+SAR | 0.502 | |
| | | | IR thermal+Thermal infrared multispectral+SAR | 0.457 | |
| | | | RGB+IR thermal+Thermal infrared multispectral+SAR | 0.500 | |
| | 50-100m | RGB、IR thermal、Thermal infrared multispectral、SAR | RGB+IR thermal | 0.526 | RGB+Thermal infrared multispectral |
| | | | RGB+Thermal infrared multispectral | 0.538 | |
| | | | RGB+SAR | 0.520 | |
| | | | IR thermal+Thermal infrared multispectral | 0.480 | |
| | | | IR thermal+SAR | 0.462 | |
| | | | Thermal infrared multispectral+SAR | 0.474 | |
| | | | RGB+IR thermal+Thermal infrared multispectral | 0.523 | |
| | | | RGB+IR thermal+SAR | 0.497 | |
| | | | IR thermal+Thermal infrared multispectral+SAR | 0.467 | |
| | | | RGB+IR thermal+Thermal infrared multispectral+SAR | 0.500 | |
| | 100-120m | Hyperspectral、Thermal infrared multispectral、SAR | Hyperspectral+Thermal infrared multispectral | 0.511 | Hyperspectral+Thermal infrared multispectral |
| | | | Hyperspectral+SAR | 0.487 | |
| | | | Thermal infrared multispectral+SAR | 0.486 | |
| | | | Hyperspectral+Thermal infrared multispectral+SAR | 0.493 | |

*(Continued)*

**Table 9.** (Continued)

| Area Type | Height | Candidate Sensor Set | Combination Scheme | Closeness | Optimal Scheme |
|---|---|---|---|---|---|
| **Rural Rolling** | 5-20m | RGB、LiDAR、SAR | RGB+LiDAR | 0.611 | RGB+LiDAR |
| | | | RGB+SAR | 0.496 | |
| | | | LiDAR+SAR | 0.389 | |
| | | | RGB+LiDAR+SAR | 0.499 | |
| | 20-50m | RGB、LiDAR、SAR | RGB+LiDAR | 0.598 | RGB+LiDAR |
| | | | RGB+SAR | 0.522 | |
| | | | LiDAR+SAR | 0.402 | |
| | | | RGB+LiDAR+SAR | 0.509 | |
| | 50-100m | RGB、LiDAR、SAR | RGB+LiDAR | 0.590 | RGB+LiDAR |
| | | | RGB+SAR | 0.548 | |
| | | | LiDAR+SAR | 0.410 | |
| | | | RGB+LiDAR+SAR | 0.518 | |
| | 100-120m | RGB、Hyperspectral、L-iDAR、SAR | RGB+Hyperspectral | 0.510 | Hyperspectral+SAR |
| | | | RGB+LiDAR | 0.472 | |
| | | | RGB+SAR | 0.495 | |
| | | | Hyperspectral+LiDAR | 0.505 | |
| | | | Hyperspectral+SAR | 0.528 | |
| | | | LiDAR+SAR | 0.490 | |
| | | | RGB+Hyperspectral+LiDAR | 0.497 | |
| | | | RGB+Hyperspectral+SAR | 0.516 | |
| | | | Hyperspectral+LiDAR+SAR | 0.507 | |
| | | | RGB+Hyperspectral+LiDAR+SAR | 0.500 | |
| **Rural Mountainous** | 5-20m | RGB、LiDAR、SAR | RGB+LiDAR | 0.562 | RGB+LiDAR |
| | | | RGB+SAR | 0.436 | |
| | | | LiDAR+SAR | 0.438 | |
| | | | RGB+LiDAR+SAR | 0.476 | |
| | 20-50m | RGB、LiDAR、SAR | RGB+LiDAR | 0.549 | RGB+LiDAR |
| | | | RGB+SAR | 0.465 | |
| | | | LiDAR+SAR | 0.451 | |
| | | | RGB+LiDAR+SAR | 0.487 | |
| | 50-100m | RGB、LiDAR、SAR | RGB+LiDAR | 0.541 | RGB+LiDAR |
| | | | RGB+SAR | 0.494 | |
| | | | LiDAR+SAR | 0.459 | |
| | | | RGB+LiDAR+SAR | 0.498 | |
| | 100-120m | LiDAR、SAR | LiDAR+SAR | - | LiDAR+SAR |
| **Urban/ suburban** | 5-20m | RGB、IR thermal、Thermal infrared multispectral | RGB+IR thermal | 0.561 | RGB+IR thermal |
| | | | RGB+Thermal infrared multispectral | 0.528 | |
| | | | IR thermal+Thermal infrared multispectral | 0.425 | |
| | | | RGB+IR thermal+Thermal infrared multispectral | 0.500 | |
| | 20-50m | RGB、IR thermal、Thermal infrared multispectral | RGB+IR thermal | 0.559 | RGB+IR thermal |
| | | | RGB+Thermal infrared multispectral | 0.530 | |
| | | | IR thermal+Thermal infrared multispectral | 0.422 | |
| | | | RGB+IR thermal+Thermal infrared multispectral | 0.499 | |
| | 50-100m | RGB、IR thermal、Thermal infrared multispectral | RGB+IR thermal | 0.540 | RGB+IR thermal |
| | | | RGB+Thermal infrared multispectral | 0.524 | |
| | | | IR thermal+Thermal infrared multispectral | 0.436 | |
| | | | RGB+IR thermal+Thermal infrared multispectral | 0.494 | |
| | 100-120m | RGB、Thermal infrared multispectral | RGB+Thermal infrared multispectral | - | RGB+Thermal infrared multispectral |

sensors is directly determined as the optimal scheme for the current area-height condition. If the number of candidate sensors is ≥ 3, all possible multi-sensor fusion schemes need to be evaluated. The specific steps are: first, construct a combined effective performance matrix. For each combination scheme, calculate its average effective performance score across all metrics. Second, perform standardization and weighting. Standardize the combined effective performance matrix using vector normalization to eliminate metric scale differences, and then calculate the weighted standardized matrix based on the metric weights determined by improved AHP. Finally, calculate the closeness of each combination scheme, rank them in descending order of closeness, and select the optimal combination. The sensor fusion scheme selection results for the four major areas at different flying heights, calculated through the above process, are shown in Table 9.

## 4.4. Optimal sensor fusion results

The optimal sensor fusion scheme selected via the TOPSIS method is shown in Table 10. As indicated in Table 10, the optimal sensor fusion exhibits a correlation between scenario and flying height due to differences in environmental adaptability and hazard response requirements across different areas. For rural level highway areas, the sensor combination progressively transitions from basic optical (RGB + IR) to more refined optical, such as RGB+Thermal infrared multispectral and Hyperspectral+Thermal infrared multispectral, as UAV flying height increases, so as to address the issue of performance attenuation with height. Rural rolling area highways primarily utilize RGB + LiDAR, with hyperspectral and SAR technology (Hyperspectral+SAR) introduced at high heights (100-120m). Rural mountainous area highways also primarily utilize RGB + LiDAR, while adjusting to LiDAR + SAR at 100-120m to better adapt to monitoring requirements. Urban/suburban highways stably employ RGB + IR at heights between 5-100m, upgrading to RGB+Thermal infrared multispectral only at 100-120m.

Meanwhile, studies in References [41,42], based on UAV field monitoring data, provide empirical support for the effectiveness of the optimal sensor fusion schemes selected in this work. In Reference [41], road segmentation experiments were conducted using RGB and LiDAR fusion based on the KITTI dataset. The results demonstrated that this fusion scheme achieved a maximum F-measure (MaxF) of 97.20%, matching the performance of the advanced PLARD model. Compared to single-modal models such as RGB (96.57%), LiDAR-ADI (96.37%), and LiDAR-ALT (95.29%), its detection accuracy was significantly enhanced. This fully validates the application potential and performance advantages of the fusion scheme in complex terrain monitoring. Reference [42], through the construction of the bimodal road crack identification dataset RoadCrack-MM-2025, validated the performance of the RGB and Infrared (IR) fusion scheme under different lighting conditions. Under weak-light conditions, this fusion scheme improved the Precision, Recall, mAP@0.5, and mAP@0.5:0.95 from 83.8%, 81.5%, 84.9%, and 41.7% to 95.3%, 90.5%, 92.9%, and 56.3%, respectively. In no-light conditions, although the RGB single modality performed poorly (with an mAP@0.5 of only 67.5%), the fusion model maintained high levels of Precision, Recall, and mAP@0.5 at 95.3%, 90.5%, and 92.9%, respectively, by incorporating IR

Table 10. Optimal sensor fusion scheme for different areas and heights.

| Area Type | Low (5–20 m) | Medium-low (20–50 m) | Medium-high (50–100 m) | High (100–120 m) |
|---|---|---|---|---|
| Rural Level | RGB + IR | RGB+Thermal infrared multispectral | RGB+Thermal infrared multispectral | Hyperspectral+Thermal infrared multispectral |
| Rural Rolling | RGB + LiDAR | RGB + LiDAR | RGB + LiDAR | Hyperspectral+SAR |
| Rural Mountainous | RGB + LiDAR | RGB + LiDAR | RGB + LiDAR | LiDAR + SAR |
| Urban/suburban | RGB + IR | RGB + IR | RGB + IR | RGB+Thermal infrared multispectral |

thermal radiation features, demonstrating exceptional adaptability to extreme environments. In summary, these validation results consistently indicate that the core schemes, such as RGB + LiDAR and RGB + IR, selected in this study through the improved AHP-TOPSIS method, possess reliable capabilities for identifying disaster precursors in practical engineering applications, fully reflecting the practical engineering value of the proposed method.

## 4.5. Discussion

The core sensor fusion schemes selected in this study, including RGB + IR and RGB + LiDAR, provide a hardware configuration basis for the high-precision monitoring of highway area hazards. To further explore the application potential of these sensor combinations and improve monitoring accuracy in complex environments, efficient data fusion algorithms can be explored for different combination types. The details are as follows:

For the RGB + LiDAR combination, a dual-modal fusion model can be constructed based on the U-Net encoder-decoder architecture, whose core fusion strategy is the early fusion of RGB and LiDAR-ADI. This method first converts LiDAR point clouds into altitude image (ALT) and ADI through projection and interpolation, which are the same size as RGB images. ADI may provide smooth intensities on roads to achieve efficient fusion with RGB visual features (its fusion adaptability is significantly better than that of ALT). Then, the RGB and LiDAR-ADI images are concatenated in the depth dimension before being input into the model for unified training, enabling collaborative detection through the deep fusion of dual-modal features. Relevant experimental verification demonstrates that compared with single sensor modalities, this fusion strategy has significant advantages in road segmentation tasks and can improve the MaxF to 97.20% [41].

For the RGB + IR combination, Cross-Modality Fusion Transformer mechanisms, such as the one employed by YOLOv11-DCFNet, can be introduced. This mechanism extracts visible light and infrared features separately through a dual-branch network and achieves deep feature fusion via cross-modality self-attention. It effectively utilizes infrared thermal radiation information to compensate for the texture loss of visible light under low-illumination conditions, thus significantly enhancing detection robustness in weak-light and no-light environments. Relevant experimental validation shows that compared with the RGB single-modal model, RGB + IR fusion can improve Precision, Recall, mAP@0.5 and mAP@0.5:0.95 by 11.5%, 9.0%, 8.0% and 14.6%, respectively; especially in no-light conditions, the mAP@0.5 can still be maintained at 92.9%, which is significantly higher than the 67.5% achieved by the single-modal model [42].

In addition, this study relies on the empirical judgments of domain experts when using the improved AHP method to determine metric weights, which may lead to certain subjective deviations. In the future, the sample range of expert evaluations can be expanded, and the professional backgrounds of experts can be broadened (e.g., covering highway engineering, remote sensing technology, geological disaster prevention and control, and other fields) to further reduce subjective bias and enhance the robustness and reliability of the decision-making of the model.

## 5. Conclusions

This study addresses the dynamic monitoring requirements for rural level, rural rolling, rural mountainous, and urban/sub-urban highway area slope instability and pavement cracking by proposing an improved AHP-TOPSIS UAV sensor fusion screening method. A quantitative correlation model between highway areas and sensor performance was constructed, providing an actionable screening process for dynamic UAV sensor configuration, which provides accurate, efficient, and highly adaptable technical support for highway hazard monitoring. This holds significant practical value for enhancing highway safety operations and management.

By transforming qualitative requirements into quantitative weights through the improved AHP method (with consistency ratios of all four scenario indicator weights satisfying CR < 0.1), and incorporating flight-height-corrected sensor performance scores, this method enables the scientific evaluation of individual sensor selection and multi-sensor fusion solutions. It ultimately outputs optimal fusion solutions for the four scenarios at different flight heights, demonstrating its the effectiveness and advantages in enhancing the accuracy of early hazard identification in highway areas.

                                       

Experimental data indicate that the optimal sensor fusion solutions for different scenarios exhibit significant scenario adaptability and strong correlation characteristics:

(1) In rural level highway scenarios, the RGB + IR combination achieves a comprehensive closeness degree of 0.549 at low heights (5–20 m). As flight height increases, the fusion scheme gradually transitions toward more refined optical combinations. At low to medium heights (5–100 m), RGB + Thermal infrared multispectral is adopted (0.545, 0.538), and at high heights (100–120 m), the scheme is upgraded to Hyperspectral + Thermal infrared multispectral (0.511).

(2) In rural rolling highway scenarios, RGB + LiDAR serves as the core combination, maintaining a stable closeness degree between 0.559 and 0.611 across the 5–100 m height range. At high heights (100–120 m), the combination shifts to leverage the advantages of Hyperspectral and SAR, adopting Hyperspectral + SAR (0.528). The fusion schemes effectively address the challenges of micro-crack and deformation monitoring under vegetation occlusion.

(3) In rural mountainous highway scenarios, RGB + LiDAR is the core combination, maintaining a closeness degree between 0.541 and 0.562 across the 5–100 m height range. At high heights (100–120 m), however, the scheme's emphasis is placed on utilizing SAR's all-weather imaging capability and LiDAR's penetration ability, selecting the LiDAR + SAR combination to meet the reliability requirements for slope and deformation monitoring.

(4) In urban/suburban highway scenarios, at low to medium heights of 5–100 m, the RGB + IR combination achieves a closeness degree ranging from 0.540 to 0.561. At high heights (100–120 m), it is upgraded to the RGB + Thermal infrared multispectral combination, which is suitable for monitoring pavement cracks and subsurface structural anomalies in environments with strong electromagnetic interference.

The core combinations identified in this study, such as RGB + LiDAR and RGB + IR, have also been validated by experimental data from existing literature, confirming the practical value of the screening results. In the future, the engineering application value of the research findings can be further enhanced by optimizing data fusion algorithms and reducing the subjective bias of metric weights.

## Acknowledgments

We thank all participants for their time and their commitment.

## Author contributions

**Conceptualization:** Hui Wu, Yi Lu, Jianbin Xie.

**Data curation:** Hui Wu, Kaiyuan Hu, Wei Zheng.

**Formal analysis:** Hui Wu, Shaopeng Li.

**Methodology:** Hui Wu.

**Project administration:** Jianbin Xie.

**Writing – original draft:** Hui Wu, Shaopeng Li.

**Writing – review & editing:** Hua Shan, Yi Lu, Jianbin Xie.

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
