## [Decision Letter · Decision Letter 0]

18 Jan 2026

PONE-D-25-48865Research on Multi-Sensor Fusion Architecture for Highway Area Hazard Monitoring Based on UAVsPLOS One

Dear Dr. WU,

Thank you for submitting your manuscript to PLOS ONE. After careful consideration, we feel that it has merit but does not fully meet PLOS ONE’s publication criteria as it currently stands. Therefore, we invite you to submit a revised version of the manuscript that addresses the points raised during the review process.

If applicable, we recommend that you deposit your laboratory protocols in protocols.io to enhance the reproducibility of your results. Protocols.io assigns your protocol its own identifier (DOI) so that it can be cited independently in the future. For instructions see: https://journals.plos.org/plosone/s/submission-guidelines#loc-laboratory-protocols. Additionally, PLOS ONE offers an option for publishing peer-reviewed Lab Protocol articles, which describe protocols hosted on protocols.io. Read more information on sharing protocols at . Additionally, PLOS ONE offers an option for publishing peer-reviewed Lab Protocol articles, which describe protocols hosted on protocols.io. Read more information on sharing protocols at https://plos.org/protocols?utm_medium=editorial-email&utm_source=authorletters&utm_campaign=protocols..

We look forward to receiving your revised manuscript.

Kind regards,

Xuecai Xu, Ph.D.

Academic Editor

PLOS One

Journal Requirements:

Additional Editor Comments:

The study focuses on five common sensors, but there are several other sensors available for UAV-based monitoring (e.g., hyperspectral, thermal infrared multispectral). A broader evaluation of sensor options would strengthen the study.

The analysis relies primarily on theoretical sensor performance metrics and flying height correction coefficients. Incorporating field data from actual UAV monitoring campaigns would provide more robust validation of the proposed method.

The paper mentions the need for exploring data fusion algorithms but does not delve into specific techniques. Discussing potential data fusion approaches and their impact on monitoring performance would be beneficial.

The study focuses on four specific highway areas. Discussing the generalizability of the results to other types of roads and regions would enhance the paper's value.

I recommend that the paper be accepted with minor revisions.

Reviewer's Responses to Questions

**Comments to the Author**

1. Is the manuscript technically sound, and do the data support the conclusions?

Reviewer #1: Yes

2. Has the statistical analysis been performed appropriately and rigorously? 

Reviewer #1: Yes

3. Have the authors made all data underlying the findings in their manuscript fully available?

Reviewer #1: Yes

4. Is the manuscript presented in an intelligible fashion and written in standard English?

Reviewer #1: Yes

5. Review Comments to the Author

Reviewer #1: Minor revisions are need in the structure of the paper

1. Objectives of the study has not been mentioned anywhere in the paper. Add a separate section on the objectives of the study clearly stating the objectives of the study along with scope and limitations.

2. Add section on literature review before materials and methods section,

3. AHP has been used in the research so add the copy of the questionnaire in the annexure,

4. Support conclusions with data from analysis and results sections.

6. PLOS authors have the option to publish the peer review history of their article (what does this mean?). If published, this will include your full peer review and any attached files.). If published, this will include your full peer review and any attached files.

.

Reviewer #1: **Yes:** Tejwant Singh BrarTejwant Singh Brar

---

## [Author Response · Author response to Decision Letter 1]

27 Feb 2026

Reviewer's Review Comments

1．Objectives of the study has not been mentioned anywhere in the paper. Add a separate section on the objectives of the study clearly stating the objectives of the study along with scope and limitations.

Response: Thank you for the valuable feedback. We have addressed this point by adding a clear and concise statement regarding the study's core objectives, application scope, and inherent limitations in the second paragraph of the Introduction section of the revised manuscript.

2．Add section on literature review before materials and methods section.

Response: Thank you for the suggestion. We have added a dedicated "Literature Review" section before the "Materials and Methods" section in accordance with your suggestion.

3．AHP has been used in the research so add the copy of the questionnaire in the annexure.

Response: Thank you for the suggestion. We have added a "Questionnaire" appendix to the paper, which is the expert evaluation questionnaire used for the Analytic Hierarchy Process (AHP) in this study.

4．Support conclusions with data from analysis and results sections.

Response: Thank you for the suggestion. We have revised the Conclusions section accordingly to ensure that all stated conclusions are directly supported by the data and results from the Analysis and Results section of the paper.

---

## [Editor Report · Decision Letter 1]

3 Mar 2026

PONE-D-25-48865R1Research on Multi-Sensor Fusion Architecture for Highway Area Hazard Monitoring Based on UAVsPLOS One

Dear Dr. WU,

Thank you for submitting your manuscript to PLOS ONE. After careful consideration, we feel that it has merit but does not fully meet PLOS ONE’s publication criteria as it currently stands. Therefore, we invite you to submit a revised version of the manuscript that addresses the points raised during the review process.

If applicable, we recommend that you deposit your laboratory protocols in protocols.io to enhance the reproducibility of your results. Protocols.io assigns your protocol its own identifier (DOI) so that it can be cited independently in the future. For instructions see: https://journals.plos.org/plosone/s/submission-guidelines#loc-laboratory-protocols. Additionally, PLOS ONE offers an option for publishing peer-reviewed Lab Protocol articles, which describe protocols hosted on protocols.io. Read more information on sharing protocols at . Additionally, PLOS ONE offers an option for publishing peer-reviewed Lab Protocol articles, which describe protocols hosted on protocols.io. Read more information on sharing protocols at https://plos.org/protocols?utm_medium=editorial-email&utm_source=authorletters&utm_campaign=protocols..

We look forward to receiving your revised manuscript.

Kind regards,

Xuecai Xu, Ph.D.

Academic Editor

PLOS One

Journal Requirements:

**Additional Editor Comments:**

The study focuses on five common sensors, but there are several other sensors available for UAV-based monitoring (e.g., hyperspectral, thermal infrared multispectral). A broader evaluation of sensor options would strengthen the study.

The analysis relies primarily on theoretical sensor performance metrics and flying height correction coefficients. Incorporating field data from actual UAV monitoring campaigns would provide more robust validation of the proposed method.

The paper mentions the need for exploring data fusion algorithms but does not delve into specific techniques. Discussing potential data fusion approaches and their impact on monitoring performance would be beneficial.

The study focuses on four specific highway areas. Discussing the generalizability of the results to other types of roads and regions would enhance the paper's value.

I recommend that the paper be accepted with minor revisions.

---

## [Editor Report · Decision Letter 2]

13 Apr 2026

Research on Multi-Sensor Fusion Architecture for Highway Area Hazard Monitoring Based on UAVs

PONE-D-25-48865R2

Dear Dr. Wu,

We’re pleased to inform you that your manuscript has been judged scientifically suitable for publication and will be formally accepted for publication once it meets all outstanding technical requirements.

An invoice will be generated when your article is formally accepted. Please note, if your institution has a publishing partnership with PLOS and your article meets the relevant criteria, all or part of your publication costs will be covered. Please make sure your user information is up-to-date by logging into Editorial Manager at Editorial Manager® and clicking the ‘Update My Information' link at the top of the page. For questions related to billing, please contact  and clicking the ‘Update My Information' link at the top of the page. For questions related to billing, please contact billing support..

Kind regards,

Xuecai Xu, Ph.D.

Academic Editor

PLOS One

Additional Editor Comments (optional):

The authors have made the revision as required, no more further questions.
---

## [Editor Report · Acceptance letter]

PONE-D-25-48865R2

PLOS One

Dear Dr. WU,

I'm pleased to inform you that your manuscript has been deemed suitable for publication in PLOS One. Congratulations! Your manuscript is now being handed over to our production team.

Kind regards,

on behalf of

Dr. Xuecai Xu

Academic Editor

PLOS One